# Hederacolchiside A1 Suppresses Autophagy by Inhibiting Cathepsin C and Reduces the Growth of Colon Cancer

**DOI:** 10.3390/cancers15041272

**Published:** 2023-02-16

**Authors:** Solbi Kim, Kyung-Ha Lee, Hui-Ji Choi, Eunji Kim, Sora Kang, Minju Han, Heung Jin Jeon, Mi-Young Yun, Gyu-Yong Song, Hyo Jin Lee

**Affiliations:** 1Department of Medical Science, Chungnam National University College of Medicine, Daejeon 34134, Republic of Korea; 2Department of Surgery, Chungnam National University College of Medicine, Daejeon 35015, Republic of Korea; 3College of Pharmacy, Chungnam National University, Daejeon 34134, Republic of Korea; 4Department of Internal Medicine, Chungnam National University College of Medicine, Daejeon 35015, Republic of Korea; 5Infection Control Convergence Research Center, Chungnam National University College of Medicine, Daejeon 34134, Republic of Korea; 6Department of Beauty Science, Kwangju Women’s University, Gwangju 62396, Republic of Korea

**Keywords:** autophagy, cathepsin C, cell cycle, colon cancer, hederacolchiside A1, patient-derived colon organoid

## Abstract

**Simple Summary:**

Autophagy plays an important role in the survival of cancer cells under stress conditions that are poor for survival. Therefore, inhibition of autophagy is considered a novel approach to treating cancer. Hederacolchiside A1 (HA1) is known to have anticancer effects, but the relationship with autophagy has not been proven. In this study, we evaluated the effect of HA1 on the inhibition of autophagy and cell growth in colon cancer and presented its potential as a therapeutic agent.

**Abstract:**

While autophagy degrades non-functional or unnecessary cellular components, producing materials for synthesizing cellular components, it can also provide energy for tumor development. Hederacolchiside A1 (HA1) derived from *anemone raddeana* has anticancer effects on several carcinomas by inducing apoptosis or exhibiting cytotoxicity, but the relationship with autophagy has not been studied. We investigated the association between HA1 and autophagy and evaluated its anticancer effect on colon cancer. HA1 induced accumulation of the autophagy-related markers LC3B and SQSTM1, with distinct vacuolar formation, unlike other autophagy inhibitors; the effects were similar to those of chloroquine. In addition, HA1 decreased the expression and proteolytic activity of lysosomal protein cathepsin C, reduced the growth of colon cancer cells *in vitro*, and inhibited tumor growth *in vivo*. It also reduced the expression of Ki-67 and cathepsin C in mouse tissues and reduced the growth of spheroids and organoids composed of cancer cells. Taken together, these results imply that HA1 regulates cell growth and autophagy and has potential as a promising therapeutic agent in colon cancer.

## 1. Introduction

Colorectal cancer is the second most common cause of mortality in both men and women and is declining in the long term, but this does not include mortality among younger people. Mortality from colorectal cancer increased from 2005 to 2009 by 1.2% per year in people younger than 50 years [1]. The risk factors for colon cancer include diet, obesity, genetics, alcohol consumption, and age [2,3]. For colon cancer chemotherapy, 5-fluorouracil and leucovorin with oxaliplatin (FOLFOX) or 5-fluorouracil and leucovorin with irrinotecan (FOLFIRI) in combination with bevacizumab are typically used [4]. However, repeated or long-term administration can induce side effects or drug resistance [3]. Therefore, a new therapeutic agent for colon cancer is needed. 

Autophagy decomposes and recycles old or damaged cells, organs, and proteins. The byproducts are used for energy production [5]. Autophagy is linked to both the development and suppression of cancer [6]. Cancer cells require nutrients and energy to survive under high-stress conditions, which autophagy can provide [6]. Therefore, the mechanisms by which cancer cells acquire resistance to chemotherapy via autophagy warrant further investigation. Many clinical trials of autophagy modulators for cancer are underway, either alone, such as chloroquine (CQ) or hydroxychloroquine (HCQ), or in combination with other anticancer drugs [7,8,9]. However, the role of autophagy in cancer is controversial. Therefore, further research on autophagy regulation is needed. 

Triterpenoid saponins are present in various plant species, such as ginseng and thorny spine [10]. Triterpenoid saponins have surfactant activity and can interact with cholesterol and phospholipids [11]. Triterpenoid saponins have anticancer, anti-inflammatory, and antioxidant effects [12,13]. Hederacolchiside A1 (HA1), a triterpenoid saponin derived from *anemone raddeana*, is known to induce inhibition of human umbilical vein endothelial cell (HUVEC) tubulogenesis by regulating the Ras/MEK/ERK pathway or inhibit parasites infecting humans [4,14]. However, studies have not yet been conducted on many solid tumors, and the relationship with autophagy is not clear. We have previously shown that a natural compound, decursin, inhibits autophagy through cathepsin C (CTSC), a lysosomal protease [15]. CTSC, also known as DPP-1, has been the focus of many studies as a target for the treatment of various diseases, including cancer [16,17,18]. In this study, we assessed the relationship of HA1 with autophagy in colon cancer to evaluate its therapeutic potential for cancer.

## 2. Materials and Methods

### 2.1. Chemicals and Reagents

Standard HA1 was purchased from Chengdu Biopurify Phytochemicals Co., Ltd. (Chengdu, China). HPLC-grade methanol and acetonitrile were purchased from Samchun Co., Ltd. (Pyengtaek, Republic of Korea). Water was purified using an ELGA purification system. Ethanol was purchased from Samchun Co., Ltd.

### 2.2. Plant Materials

*A. raddeana* was provided by the Global Cancer Treatment Support Foundation in Korea and was authenticated by Professor M. K. Na (College of Pharmacy, Chungnam National University).

### 2.3. Preparation of HA1

Dried roots of *A. raddeana* (1.0 kg) were ground into powder. The powder was extracted three times by reflux with methanol for 6 h and filtered. The extracts were cooled and combined, and the solvent was removed using a rotary evaporator under vacuum to yield a brownish powder (110 g). The brownish powder (110 g) was dissolved in methanol and loaded onto Diaion HP-20 resin, eluting with a gradient of aqueous ethanol (0%, 30%, 40%, 50%, 60%, 70%, and 95%) to yield major fractions (R1–R5) using thin layer chromatography (TLC) (5:1:1 butanol:acetic acid:water) compared with standard HA1. The 70% ethanol fraction (R3, 3.9 g) contained the most HA1 and was applied to a C18 resin (Cosmosil 75C_18_-PREP) and eluted with 1:1 to 2.75:1 methanol:water, yielding purified HA1 (825 mg). The structure of isolated HA1 determined with LC-MS/MS and NMR spectroscopy was identical to that of the reference [19,20]. Separation was achieved with linear–gradient elution of the two combined eluents: A (acetonitrile) and B (0.1% phosphoric acid (*v*/*v*) in water). The detailed gradient elution was: 0–28 min, 23–36% A; 28–42 min, 36–56% A; 42–52 min, 56–90% A; 52–54 min, 90–90% A; 54–58 min, 90–23% A; and 58–67 min, 23–23% A. The detection wavelength was 203 nm, and the flow rate of the mobile phase was set to 1.0 mL per minute. In addition, the concentration of HA1, which is not otherwise specified, was set to 10 μM (DMSO 0.1%).

### 2.4. Cell Culture

Human cancer cell lines and murine CT26 colon cancer cells were purchased from the American Type Culture Collection (Gaithersburg, MD) and Korean Cell Line Bank (Seoul, Republic of Korea). SW480 and HT-29 (colorectal cancer), NCI-N87 (gastric cancer), and SNU-1041 (head and neck cancer) were cultured with RPMI-1640 medium. Caki-2 (renal cancer), G-361 (melanoma), MCF-7 (breast cancer), HepG2 (liver cancer), and CT26 (colon cancer) were cultured in Dulbecco’s Modified Eagle’s Medium (DMEM; Welgene, Republic of Korea) at 37 °C in a humidified atmosphere with 5% CO_2_. 

### 2.5. Cell Proliferation Assay

In previous studies, the Cell Counting Kit-8 (CCK-8) assay has been described [21]. After 10 μL of CCK-8 solution was added to a culture medium to measure cell proliferation, incubation was performed for about 1 to 3 h, and absorbance was measured at 450nm. For the clonogenic assay, cells treated with HA1 for 24 h were washed with PBS and stained through crystal violet.

### 2.6. Western Blotting

Cells were washed twice in cold PBS and dissolved using ProEX™ CETi Lysis Buffer (Translab, Daejeon, Republic of Korea) containing protease and phosphatase inhibitors (GenDEPOT, TX) to isolate cell lysates. Cell lysates were separated with SDS-PAGE and transferred to a polyvinylidene difluoride membrane. Antibodies against LC3B-I/II (Santa Cruz Biotechnology, Santa Cruz, CA, USA; 2775), SQSTM1 (Cell Signaling Technology, Danvers, MA, USA; 39749), GAPDH (Cell Signaling Technology; 2118), and cathepsin C (CTSC; Santa Cruz Biotechnology; 74590) were used. Numbers represent relative quantified signal intensities averaged from 3 experiments. Original Western Blot figures shown in Appendix A.

### 2.7. Immunocytochemistry/Immunohistochemistry

Immunocytochemistry and immunohistochemistry were conducted as described previously [21]. Sections (4 μm thick) of paraffin-embedded tissues were prepared for staining. Slides were deparaffinized in xylene and rehydrated in a graded alcohol series. Antigen retrieval was performed using sodium citrate buffer (pH 6.0) for 30 min by microwaving, followed by blocking using blocking solution (DAKO, Denmark). The slides were incubated with each primary antibody (CTSC; Santa Cruz Biotechnology; 74590, Ki-67; Abcam; ab16667) overnight at 4 °C, washed with TBS-T, and incubated with the fluorescence-conjugated secondary antibody for 30 min. DAPI staining was used to determine the number of nuclei.

### 2.8. Transmission Electron Microscopy

Cells were treated with DMSO or 10 μM HA1 for 24 h, washed three times in PBS, and fixed in 2.5% glutaraldehyde in 0.1 M phosphate buffer (pH 7.3) overnight at 4 °C. Afterward, samples for TEM were produced from the cells at the Korea Basic Science Institute. TEM (Libra 120, Carl Zeiss, White Plains, NY, USA) was performed by the Center for Research Facilities of Chungbuk National University.

### 2.9. Proteolytic Assay

Recombinant human cathepsin C and L (rhCTSC and rhCTSL, respectively) were purchased from R&D Systems (Minneapolis, MN) and activated according to the manufacturer’s instructions. Using an activation buffer, rhCTSC and rhCTSL were diluted to final concentrations of 100 and 20 μg/mL, respectively. Activated rhCTSC was diluted to 0.5 ng/μL in the black buffer and reacted in the black plate with Gly-Arg-AMD (Bachem America, Inc., Torrance, CA), a substrate of CTSC. The HA1 treatment group was incubated together with HA1 and rhCTSC for 1 h and then reacted with the substrate.

### 2.10. siRNA

Cells were transfected with siRNA specific to CTSC (5′-GGAGAAAGUUCAUGGUAUCA-3′) using Lipofectamine RNAiMax (Invitrogen, Middlesex, MA, USA).

### 2.11. Cell Cycle Assay

Colon cancer cells were treated with HA1 for 24 h, fixed in 70% cold ethanol, and washed with PBS. Fixed cells were incubated with FxCycle™ Propidium Iodide/RNase Staining Solution (Invitrogen) for 30 min at RT. The cell cycle was analyzed using the MoFlo Cell Sorter (Kaluza 2.1, Beckman Coulter, Brea, CA, USA) and Kaluza software (ver. 1.2, Beckman Coulter).

### 2.12. Colon Cancer Spheroid Formation

Cells were seeded in ultra-low-attachment 96-well plates (Corning Inc., Corning, NY, USA) as described previously [15]. DMEM/F12 serum-free medium supplemented with 1× HEPES, 1× penicillin/streptomycin, and epidermal growth factor (10 ng/mL) was used for spheroid formation. The generation of spheroids was monitored for 2 weeks, and their diameters were measured. Cell cultures were analyzed using the Olympus CKX41 bright-field inverted microscope equipped with the eXcope XCAM1080 camera (K-OPTIC, Seoul, Republic of Korea). 

### 2.13. Allograft Mouse Model

Female BALB/c mice (6 weeks old) were maintained in an environment without specific pathogens and treated in accordance with the approval and guidelines of the Animal Experiment Ethics Committee of Chungnam National University. After CT26 was washed three times with PBS, 1 × 10^6^ cells were subcutaneously injected into the right flank of each mouse. When the tumor volume reached approximately 100 mm^3^, 100 μL HA1 was injected into the tumors twice a week. A total of 100 μL of 0.1% DMSO was used as the negative control.

### 2.14. Patient-Derived Colon Cancer Organoids

Human colon cancer and paired adjacent non-tumor tissues were collected from Chungnam National University Hospital. Colon cancer organoids were established using a similar method described previously for stomach cancer organoids [15]. Tissues were washed about 30 times with cold PBS until impurities were removed and then dissociated into single cells at 37 °C for 1 h using a Human Tumor Dissociation Kit (Miltenyi Biotec, Bergisch Gladbach, Germany). The dissociated cells were filtered using a 70-μm (BD Biosciences, San Jose, CA, USA) strainer, and unnecessary red blood cells were removed using ACK lysis buffer. The cells were resuspended in 30 μL of growth factor-reduced matrigel (Corning, Inc.) and seeded in 4-well dishes. Matrigel domes were solidified with incubation at 37 °C for 15 min and supplemented with 500 μL of complete media. Media were refreshed at 2–3 day intervals, and organoids were passaged every 2 weeks at a split ratio of 1:4-5. The Table 1 below shows the composition of the colon cancer organoid medium. 

### 2.15. Gene Expression Profiling Interactive Analysis (GEPIA)

Differential expression of CTSC in various tissues was assessed by GEPIA (Gene Expression Profiling Interactive Analysis; http://gepia.cancer-pku.cn). The GEPIA database containing 275 colon adenocarcinoma tissues and 41 normal colon tissues was downloaded for analysis. Our analysis was conducted on 10 November 2022. 

### 2.16. Statistical Analysis

Quantitative data are expressed as means ± standard deviation from at least triplicate independent experiments. The significance of differences between groups was calculated using SPSS statistics. Two-tailed Student’s t-test and Mann–Whitney U test were used where appropriate. *p* < 0.05 was considered to indicate statistical significance.

## 3. Results

### 3.1. HA1 Induces Vacuolization in Colorectal Cancer Cells

Triterpenoid saponins reportedly inhibit cancer [13,22]. HA1 has a molecular weight of 897.1 g/mol, and its structure is shown in Figure 1A. Using TLC, we obtained five major fractions (R1–R5) of a brown powder extract of A. raddeana and compared R3 with the HA1 standard (Figure 1B). The isolated and purified HA1 was compared to the standard HA1 by HPLC (Figure 1C). HA1 induced marked cytoplasmic vacuolization in colorectal cancer cell lines (Figure 1D), as well as in other cancer cells (Appendix A). 

### 3.2. HA1 Alters Autophagy

HA1 induced significant cytoplasmic vacuolization and based on this result, the effect of HA1 on autophagy was investigated. HA1 markedly increased the LC3B-II level and induced the accumulation of SQSTM1 (also known as p62), a marker of autophagy (Figure 2A). Therefore, HA1-induced SQSTM1 accumulation implies a decrease in autophagic flux. The HA1-mediated increases in the number of LC3 puncta and abnormal organelle morphology suggest dysfunctional autophagy (Figure 2B), which was recapitulated in other cell lines using Western blot (Appendix A). Unlike other autophagy inhibitors, HA1 induced vacuole formation (Appendix A). In addition, HA1 increased the LC3B-II and SQSTM1 levels similar to those induced by CQ when each inhibitor was used at a concentration and time commonly used in colorectal cancer (Appendix A). Control cells displayed normal mitochondria and other organelles, whereas HA1-treated cells exhibited autophagic vacuoles and abnormal organelles (Figure 2C). These results suggest that HA1 promotes LC3B and SQSTM1 accumulation, impacting autophagic flux.

### 3.3. HA1 Reduces Cathepsin C Expression and Activity

CTSC expression was higher in the colon cancer samples than in normal samples (Figure 3A). In addition, it was confirmed that CTSC was hardly expressed in normal cells, but significantly expressed in colon cancer cell lines (Appendix A). HA1 reduced the expression of CTSC (Figure 3B). Application of MG132, a proteasome degradation inhibitor, together with HA1 showed that the HA1-mediated reduction in CTSC expression was independent of proteasome degradation (Figure 3C). In addition, when the mRNA level of CTSC after treatment with HA1 was evaluated, it was confirmed that HA1 did not affect the transcription of CTSC (Appendix A). siRNA targeting CTSC had the same effect on autophagy markers as that of HA1 (Figure 3D). To assess the hypothesis that HA1 regulates autophagy through CTSC, CCD-18Co, a cell line that was isolated from the normal colon tissue and hardly expressed CTSC, were treated with HA1 along with other cancer cells, and autophagy markers were evaluated (Appendix A). As expected, there was no change in autophagy markers in CCD-18Co cells, but changes in the autophagy markers were induced in cancer cells with CTSC expression. Consistent with these results, public data showed that CTSC expression was higher in cancer tissues than in non-cancerous tissues (Appendix A). Furthermore, HA1 reduced not only the expression of CTSC but also its protease activity (Figure 3E). 

### 3.4. CTSC Regulates Cell Proliferation

Next, we assessed the effect of CTSC on the growth of colorectal cancer cells. A decrease in CTSC expression reduced the growth of colorectal cells and caused cell cycle arrest (Figure 4A,B). We observed that G0/G1 phase decreased in all three cell lines after treatment with siCTSC. A decrease in CTSC expression reduced spheroid viability after 5 days in colon cancer cells (Figure 4C). Therefore, decreased CTSC expression causes an unfavorable environment for the survival of colorectal cancer cells. 

### 3.5. HA1 Inhibits Cell Proliferation and Induces Cell Cycle Arrest

HA1 significantly reduced cell proliferation in a dose-dependent manner (Figure 5A), an effect confirmed by colony formation assay (Figure 5B). Propidium iodide staining revealed that HA1 significantly reduced the proportion of cells in the G0/G1 phase and increased that of cells in the S and G2/M phases, indicating that HA1 induced S and G2/M phase arrest (Figure 5C). 

### 3.6. HA1 Suppresses Tumor Growth in a CT26 Tumor Allograft Mouse Model 

A subcutaneous mouse model was used to assess the effect of HA1 on tumor growth in vivo. In contrast to control mice, HA1-treated mice showed retarded tumor growth (Figure 6A) and smaller tumors (Figure 6B,D). Body weight was not affected by HA1 treatment (Figure 6C). Histological analysis of allograft mouse tumors revealed significant differences between control tumors and HA1-treated tumors (Figure 6E). CTSC and proliferation marker Ki-67 levels were lower in HA1-treated tumors, consistent with the in vitro findings.

### 3.7. HA1 Reduces the Growth of 3D Spheroids and Patient-Derived Colon Cancer Organoids 

Control spheroids grew and had a well-formed shape with a smooth boundary, but HA1-treated spheroids were smaller and amorphous (Figure 7A,B). We established colon organoids using tissues from patients with colon cancer who underwent surgery. Histological images of tumors and adjacent non-tumor colon tissue and characteristics of the patients are described in Appendix A and Appendix A. Importantly, HA1 reduced the size and number of organoids (Figure 7C). Therefore, the effects of HA1 on tumor spheroids and patient-derived colon cancer organoids were reconfirmed, consistent with previous in vitro and in vivo findings.

## 4. Discussion

We demonstrated that HA1 suppresses autophagy and regulates the cell cycle by inhibiting CTSC in colorectal cancer. HA1 induced intracellular vacuolization and accumulation of LC3B-II and SQSTM1. In addition, it promoted the formation of abnormal intracellular organelles such as cytoplasmic vacuoles and autophagosome accumulation. Moreover, HA1 reduced the expression of CTSC, which is high in colorectal cancer and other carcinomas, thereby inhibiting cell growth and causing cell cycle arrest. Our findings indicate that HA1 might have therapeutic potential against colorectal cancer. 

Autophagy degrades cell components or organelles to provide energy [23]. Autophagy involves the formation of autophagosomes, which deliver engulfed intracellular components to lysosomes for degradation, and the lysosomes then fuse with autolysosomes [24]. The intracellular components are degraded by lysosomal proteases, a process termed autophagic flux [23,24]. The relationship between autophagy and cancer is unclear, despite intensive investigation. However, efforts to obtain anticancer effects by suppressing autophagy continue [25]. In this study, HA1 inhibited autophagy by triggering the accumulation of LC-3-II and SQSTM1, thereby inhibiting cancer survival because cancer cells tend to be more dependent on autophagy compared with normal cells [26,27,28,29]. HCQ and CQ inhibit lysosomal degradation, the final step of autophagy [7,9]. HCQ or CQ in combination with other chemotherapy drugs demonstrated therapeutic potential for cancer in multiple clinical trials [30,31,32]. HA1 induced vacuole formation in a similar manner to that induced by CQ. Combination therapy with the autophagy inhibitor 3-MA and trastuzumab showed a better effect in HER2-positive breast cancer cells, and inhibition of the autophagy-related factors ATG5, ATG7, and Beclin1 reduced resistance to tamoxifen in ER-positive breast cancer [33,34]. Therefore, autophagy is linked to cancer [35]. However, the effects of autophagy in tumor development are ambivalent and more research is needed to fully understand the role of autophagy. 

Saponins are divided into sterol saponins and triterpenoid saponins. The precursors are identical, but triterpenoid saponins have three more methyl groups than sterol saponins [36]. Triterpenoids have anti-insecticide, anti-inflammatory, and anti-diabetic activities [37,38,39]. HA1 exhibited anti-tumor activity in vitro and in vivo. HA1 induced membrane damage and cytoplasmic vacuolization in melanoma and apoptosis in HL-60 human leukemia cells [40]. In addition, HA1 affects HUVEC network formation by activating the Ras/MEK/ERK cascade and reduces HUVEC tolerance to apoptosis [4]. However, the relationship between HA1 and autophagy is unclear. Our data suggest that HA1 induces autophagy dysfunction and cell cycle arrest in colon cancer. Therefore, HA1 may have therapeutic potential by inhibiting the growth of tumors with high levels of autophagy.

HA1 induced dysregulation of autophagy by reducing the level of CTSC, a lysosomal protease. CTSCs are involved in numerous diseases, including chronic inflammation and autoimmune diseases, and their expression and activity are increased in the carcinogenesis of islet cells, skin, and mammary glands [41,42,43]. Indeed, decreased levels of CTSC reduced gastric cancer cell growth and induced cell cycle arrest, thereby inhibiting cancer progression [15]. 

Cancer stem cells increase drug resistance, and 2D culture does not mimic the in vivo environment, which explains the failure of many candidate drugs in clinical practice [44]. In cancer research, 3D culture is a good preclinical model because it mimics the microenvironment of the organ [45,46]. In this study, we evaluated the anticancer efficacy of HA1 using spheroids and patient-derived colon cancer organoids. HA1 also reduced colon cancer growth in 3D culture models, suggesting that it has therapeutic potential for colon cancer. 

## 5. Conclusions

This study revealed the importance of a novel mechanism of autophagy inhibition through CTSC reduction and cell-growth inhibition using HA1. HA1 may be a potential therapeutic candidate for the treatment of colon cancer, and further studies on the safety and efficacy of the compound are needed.

## Figures and Tables

**Figure 1 cancers-15-01272-f001:**
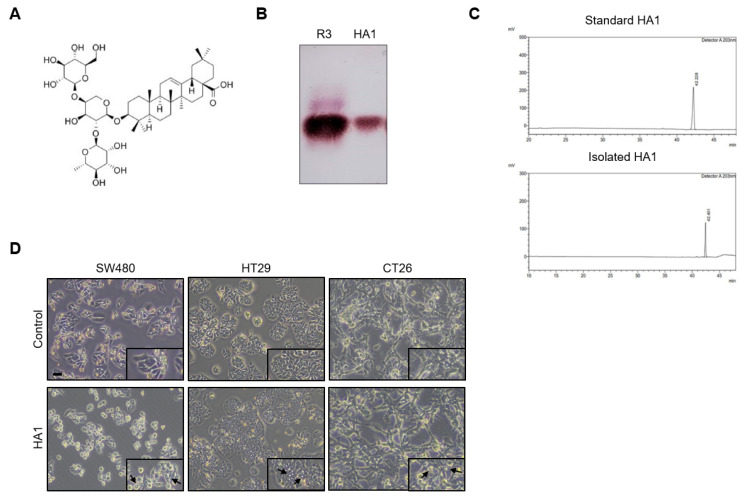
HA1 induces cellular vacuolization. (**A**) Chemical structure of HA1. The structure of HA1 was drawn using ChemDraw Professional 16.0. (**B**) TLC pattern of the R3 fraction and standard HA1 (5:1:1 butanol:acetic acid:water). (**C**) HPL chromatogram of standard HA1 and isolated HA1. (**D**) Microscopy images (×200) of human colon cancer cells treated with 10 μM HA1 for 24 h. Scale bar: 20 μm. Inset: higher magnification view of the upper image (×400). Some vacuoles are indicated by black arrows.

**Figure 2 cancers-15-01272-f002:**
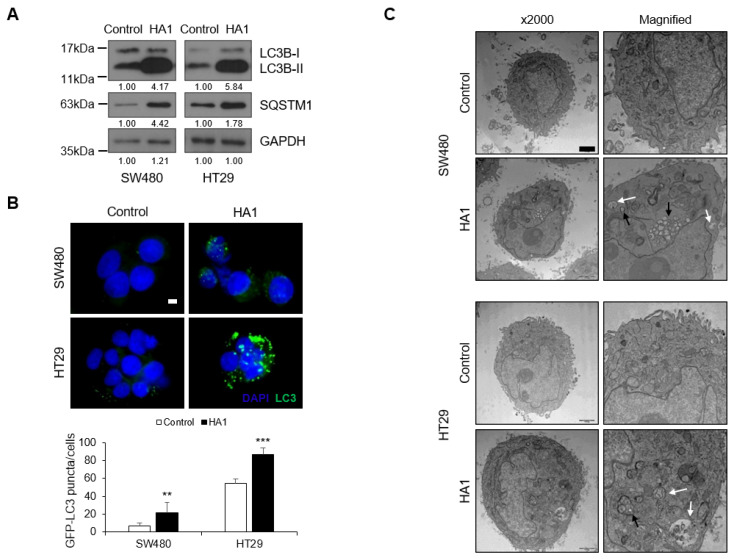
HA1 alters autophagy in colon cancer cells. (**A**) Western blotting and Image J analysis showing that HA1 increased the LC3B-II and SQSTM1 levels in SW480 and HT29 cells (n = 3). (**B**) SW480 and HT29 cell lines transfected with a GFP-LC3 plasmid and treated with HA1 for 24 h (×1000). Scale bar: 2 μm. ** *p* < 0.01; *** *p* < 0.001. (**C**) Effect of HA1 on colon cancer cell morphology. Representative TEM images are shown (×2000). Scale bar: 2 μm. Right panel: higher magnification view of the left region. White arrow: vacuole; black arrow: autophagosome.

**Figure 3 cancers-15-01272-f003:**
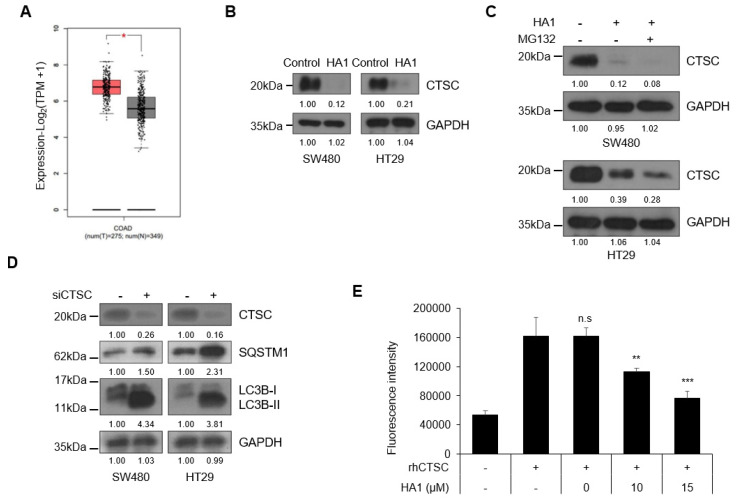
HA1 reduces CTSC expression and activity. (**A**) High expression of CTSC in colon adenocarcinoma (GEPIA http://gepia.cancer-pku.cn/). CTSC expression in normal (gray box) and cancer (red box) tissues. (**B**) Reduced CTSC expression by HA1. Western blot bands were measured using Image J software (n = 3). (**C**) No effect of MG132 (10 μM) on the HA1-mediated reduction in CTSC expression. Western blot bands were measured using Image J software (n = 3). (**D**) Autophagic flux evaluated by Western blotting after transfection with CTSC siRNA. Western blot bands were measured using Image J software (n = 3). (**E**) Proteolysis assay to measure inhibition of CTSC by HA1 using the CTSC substrate Gly-Arg-AMC. n.s.: not significant; * *p* < 0.05; ** *p* < 0.01; *** *p* < 0.001.

**Figure 4 cancers-15-01272-f004:**
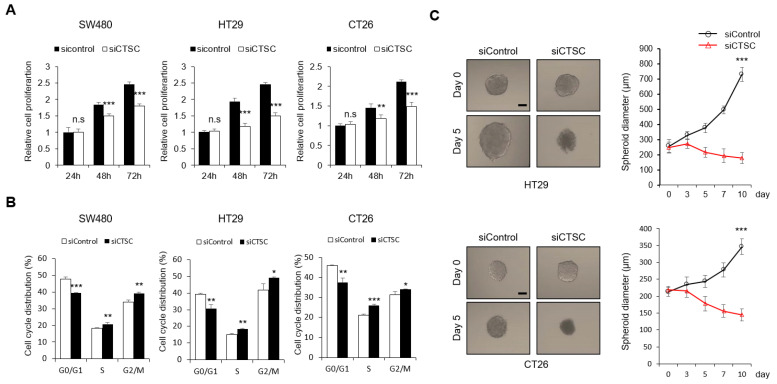
CTSC regulates cell proliferation. (**A**,**B**) Cell proliferation and cell cycle assays performed after transfection with an siRNA targeting CTSC in three cell lines. After 24 h, cells were cultured in 96-well plates. Proliferation was measured with CCK-8 assay at the indicated times. Cell cycle analysis using flow cytometry was performed at 48 h after transfection. (**C**) Spheroid formation in cells cultured in a low-attachment 6-well plate at 24 h after CTSC siRNA transfection. After the diameter of the spheroids reached approximately 200 μm, spheroids were transferred to a low-attachment 96-well plate at one spheroid per well. Bright-field images of colon cancer spheroids at 0 and 5 days after treatment with HA1 (×100). Scale bar: 100 μm. n.s., not significant; * *p* < 0.05; ** *p* < 0.01; *** *p* < 0.001.

**Figure 5 cancers-15-01272-f005:**
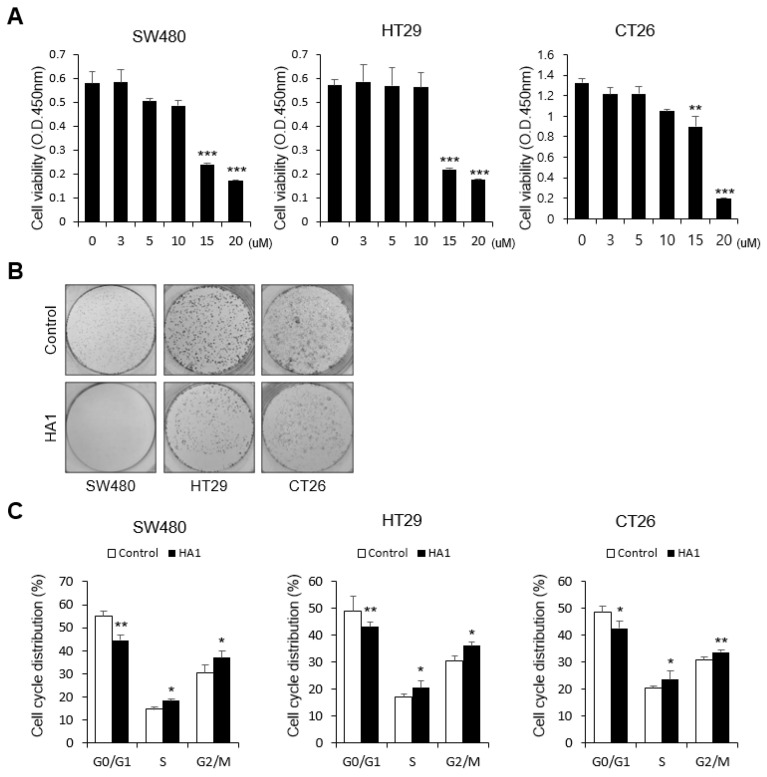
HA1 inhibits cell viability and induces cell cycle arrest. (**A**) Cell viability of SW480, HT29, and CT26 cells treated with HA1 for 24 h. (**B**) Reduced cell growth in bright-field images. (**C**) Cell cycle analysis using flow cytometry using propidium iodide (PI) staining showing an increase in the population of HA1-treated cells in the G2/M phase. The concentration of HA1 was 10 μM. * *p* < 0.05; ** *p* < 0.01; *** *p* < 0.001.

**Figure 6 cancers-15-01272-f006:**
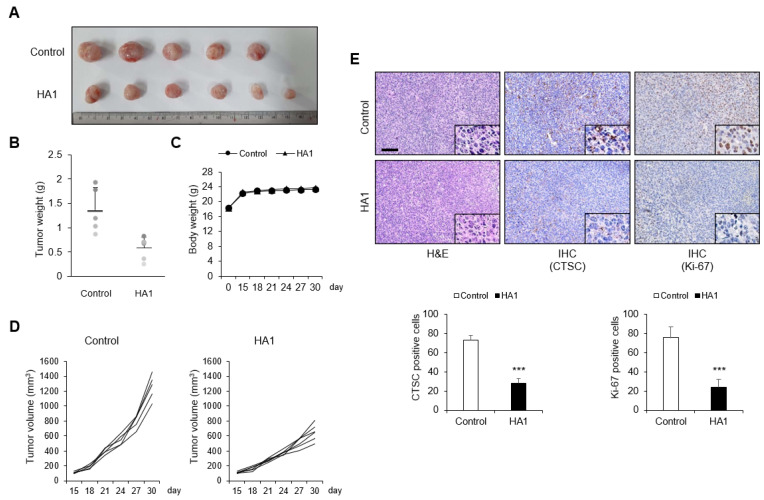
HA1 suppresses tumor growth in a CT26 tumor allograft mouse model. (**A**) Tissues dissected from a BALB/c mouse. Tumors were smaller in HA1-injected mice than control mice. (**B**) Tumor weight at study termination. (**C**) Body weight measured twice per week. (**D**) Tumor growth in tumor-bearing mice during treatment. Tumor volume (control mice: n = 5; HA1-treated mice: n = 6) was measured twice per week. Tumor volume was calculated using the formula (length × width^2^)/2. (**E**) Representative images of hematoxylin and eosin staining to demonstrate tumor histology and of CTSC and Ki-67 immunohistochemical staining (×200) and the number of CTSC and Ki-67-expressing cells. Inset: higher-magnification view of the upper image. *** *p* < 0.001.

**Figure 7 cancers-15-01272-f007:**
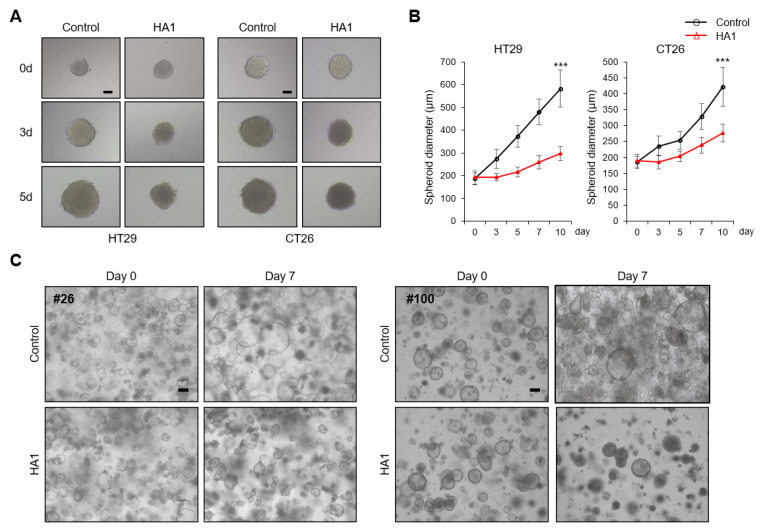
HA1 reduces the growth of patient-derived colon cancer organoids. (**A**) HA1-treated spheroids of appropriate size after transfer to a low-attachment 96-well plate (×100). Scale bar: 100 μm. (**B**) Average spheroid size from days 0 to 10. (**C**) Organoids of an appropriate size grown in a 4-well dish were treated with DMSO (control) or HA1 (10 μM) and observed for 7 days (×100). Scale bar: 100 μm. *** *p* < 0.001.

**Table 1 cancers-15-01272-t001:** Composition of the colon cancer organoid medium.

Supplement	Stock con.	Final con.	Catalog #	Company
Advanced DMEM/F12	-	-	12634-010	Gibco
HEPES	1 M	10 mM	15630-106	Gibco
Penicilin/streptomycin	100X	1X	SV30010	Hyclone
B27	50X	1X	17504044	Gibco
N2	100X	1X	17502048	Gibco
GlutaMax	100X	1X	A12860-01	Gibco
EGF	100 μg/mL	50 ng/mL	AF-100-15	Peprotech
Noggin	100 μg/mL	100 ng/mL	250-38	Peprotech
R-spondin 1	100%	10%	3710-001-K	Trevigen
Gastric I	10 μM	10 nM	G9145	Sigma
FGF-10	10 μg/mL	100 ng/mL	100-26	Peprotech
Primocin	500X	1X	ant-pm-1	Invivogen
Wnt-3A	100%	50%	CRL-2647	ATCC
Y-27632	10 mM	10 μM	Y0503	Sigma
Nicotinamide	1 M	10 mM	N0636	Sigma
SB202190	20 mM	10 μM	S7067	Sigma
A83-01	5 mM	0.5 μM	2939	Tocris
N-acetylcysteine	1 M	1 mM	A9165	Sigma
Prostaglandin E2	1 mM	500 nM	2296	Tocris

# means catalog number.

## Data Availability

The data presented in this study are contained within the article and the Appendix A.

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
