# Peer review of "Hederacolchiside A1 Suppresses Autophagy by Inhibiting Cathepsin C and Reduces the Growth of Colon Cancer"

_cancers, 2023, doi:10.3390/cancers15041272_

Round 1

Reviewer 1 Report (Previous Reviewer 1)

The resubmit manuscript is much better than before.

The authors reviesed it to reflect all the requirements.

Now, ready for published.

Author Response

Your comments made our manuscript much better. We are very grateful to you. 

Reviewer 2 Report (Previous Reviewer 3)

The manuscript was profoundly revised and corrected by the authors. Of the detailed response letter and the corrections highlighted in the manuscript, all my concerns were successfully addressed.

I liked in particular the new table in chapter 2.14, which, alongside with scientific soundness, might benefit as source of citations in the future.

Small changes suggested:

line106: "colon cancer" for CT26 should be indicated.

line 202: only the software is indicated, please specify the assays used (e.g. for normally distributed data, Student's t-test was used, for cell cycle in vivo data, Mann-Whitney test (or other test(s) ) was/were used.

line 224: there was a misunderstanding regarding my previous comment regarding SF4B (as I see, now it is SF5B):

"what does red color mean on the top legend" - I pointed that the very small cancer type names above the graph are colored black or red. Are they referring to significant differences?

furthermore, by big magnification I could read that in the bottom axis, T/N and (number of samples) are indicated. While it is correct, I suggest to remove that information since it is not readable. If the data are mined from a public database that must be properly referenced and mentioned in the methods section. For this, I think date of data access is a more reliable parameter than number of samples involved, as public databases can grow and change over time.

Figure 3A: if the y axis shows raw absorbance, than it cannot be named cell proliferation, as the latter is usually percentage or ratio to the untreated cells. (in this case, the problem of proliferaRtion is also solved). Same issue for fig5A y-axes.

Line 299: my previous question was about: since cells do not divide much in 24 hours, cell viability is much closer to what the measurement reflects (still not appropriate to name y axes after that).

I believe that my comments require only formal changes in the otherwise valuable manuscript.

Author Response

Reviwer #2

The manuscript was profoundly revised and corrected by the authors. Of the detailed response letter and the corrections highlighted in the manuscript, all my concerns were successfully addressed.

I liked in particular the new table in chapter 2.14, which, alongside with scientific soundness, might benefit as source of citations in the future.

First, thank you for your informative comment which made our manuscript much better.

Small changed suggested:

Line106: “colon cancer” for CT26 should be indicated.

We have added.

Line 202: only the sofeware is indicated, please specify the assays used (e.g. for normally distrubuted data, Student’s t-test was used, for cell cycle in vivo data, Mann-Whitney test (or other test(s)) was/were used.  

We have specified.

Line 224: there was a misunderstanding regarding my previous comment regarding SF4B (as I see, now it is SF5B)

“what does red color mean on the top legend” – I pointed that the very small cancer type names above the graph are colored black or red. Are they referring to significant differences?

Furthermore, by big magnification I could read that in the bottom axis, T/N and (number of samples) are indicated. While it is correct, I suggest to remove that information since it is not readable. If the data are mined from a public database that must be properly referenced and mentioned in the methods section. For this, I think data of data access is a more reliable parameter than number of samples involved, as public databased can grow and change over time.

We apologize for any misunderstanding and fully agree with your opinion. First of all, we have removed the unreadable indication on the bottom axis. Regarding black and red small cancer type names above the graph, CTSC expression in red colored cancer types was upregulated. Thus, we have added appropriate description in figure legend. Also, according to your comment, we have added detailed description of public data.

Figure 4A: if the y axis shows raw absorbance, than it cannot be named cell proliferation, as the latter is usually percentage or ratio to the untreated cells. (in this case, the problem of proliferation is also solved). Same issus for fig5A-y-axis.

We have renamed it as a relative value compared to untreated cells following your comment.

Line 299: my previous question was about: since cells do not divide much in 24 hours, cell viability is much closer to what the measurement reflects (still not appropriate to name y axes after that).

We agree with you. We have replaced cell proliferation with cell viability.

Reviewer 3 Report (Previous Reviewer 4)

Dear authors,

I would like to thank you for your kind response. In my opinion, you managed to answer at the majority of my questions, however,  there are 3 points that I need more details.

Point 1: Many cell lines were added but you presented results for 3 cell lines.

Point 2: Related to LC3, I think that the problem is not only that you did not present the right antibody but LC3s (A, B, C) may present different functions. 

Point 8: I undestand that you prsent the proliferation but I still believe that the y-axis is a bit confused. I mean, you did not refer the values 1, 2, 3 etc what exactly respresent. My suggestion would be to present a relative proliferation comparing samples versus contol.

Kind regards

Author Response

I would like to thank you for your kind response. In my opinion, you managed to answer at the majority of my questions, however, there are 3 points that I need more details.

First, thank you for your valuable comment which made our manuscript much better.

  1. Many cell lines were added but you presented results for 3 cell lines.

In our work, we assessed the effect of HA1 in colorectal cancer. So we presented the results of 3 colon cancer cell lines.

We also evaluated the effect of HA1 on several cancers to see whether HA1 induce similar effects or not. Then we added the results in the supplementary dataset.

  1. Related to LC3, I think that the problem is not only that you did not present the right antibody but LC3s (A, B, C) may present different functions.

We apologize for any inaccurate information. We used the LC3B antibody in this experiment. In many studies, LC3B was used to assess changes caused by autophagy, and we  selected LC3B and used it in our study. All notations related to this have been corrected.

  1. I undestand that you present the proliferation but I still believe that the y-axis is a bit confused. I mean, you did not refer the values 1, 2, 3 etc what exactly respresent. My suggestion would be to present a relative proliferation comparing samples versus contol.

We have renamed it as a relative value compared to untreated cells following your comment.

This manuscript is a resubmission of an earlier submission. The following is a list of the peer review reports and author responses from that submission.

Round 1

Reviewer 1 Report

In this manuscript, Solbi Kim et al showed that  Hederacolchiside A1 suppresses autophagy by inhibiting cathepsin C and reduces the growth of colon cancer. These findings are potentially interesting. The manuscript could be further strengthened with a few additional experiments denoted below.

1. The authors need to explain more about Hederacolchiside A1 (include Triterpenoid saponins) in introduction part.

2. The reason for this experiment and the connection is weak. Please supplement your opinion in introduction part.

3. There is no explanation as to how the CTSC was found.

4. There are insufficient related experiments on whether HA regulates the autophagy by regulating the CTSC.

5. There are several places that incorrectly or inaccurately write down the manuscript. Authors need to pay close attention to proper labeling of this manuscript.

Author Response

First, thank you for your informative comments. I hope the revised manuscript

is now suitable for publication in the Cancers. We look forward to your favorable reply.

Sincerely, 

Reviewer 2 Report

A brief summary

The authors purified Hederaccolchiside A1 (HA1) from anemone raddeana and demonstrated the suppressive effect of autophagy using colorectal cancer cells. The authors found that HA1 treatment increases LC3-II and SQSTN1 which are autophagy markers, however, HA1-treatment causes abnormal vacuoles like an incomplete form of autophagic vacuoles. The authors demonstrated that HA1 treatment inhibits cathepsin C in both expression and protease activity, resulting in the arrest of the cell cycle. The authors also performed a xenograft mouse model and patient-derived colon cancer organoids and clearly proved that HA1 suppresses tumor growth. These studies are generally well done.

General concept comments

1.    The suppressive effect of autophagy against cancer cells is very clear in Figure1, Supplemental Figure1, and supplemental Figure 2. It would be good to mention the effect of HA on normal cells in the manuscript. I wonder if normal cells treated with HA decrease autophagy. If so, is the suppressive effect in cancer cells stronger than in normal cells? 

2.    The functional mechanism of HA to decrease Cathepsin C in both expression and protease activity is not clear. It would be good in the manuscript. 

3.  In figure 6, it is better to mention how the author decides the dose amount of HA1. I wonder if the amount of HA1 is increased or the dosing period is extended, it is possible to suppress the tumor growth more.  

4.    In figure 6, it is also better to mention side effects.

Author Response

(The authors gave the same response as above.)

Reviewer 3 Report

The authors investigated the role of HA-1 on colon cancer growth and processes, highlightedly autophagy. Since the general role of the compound is already known, the novelty is aimed to connect the effects with autophagy suppression. The manuscript is well written, with some parts to be improved.

In part, the manuscript gives a nice clear logical model affecting CTSC and altering autophagy flux marked by LC3-II and SQSTM1. This part has novelty and a good message.

On the other hand, many other things, unrelated to autophagy, are difficult to follow. As proliferation was already cited in the introduction, it was still assayed on multiple figures. Proliferation and viability should be clearly used.

Vacuole formation can be interesting, but not really put in context. Some autophagy inhibitors did not seem to have autophagy-inhibitory effect at all. Electronmicroscopy images and organelle integrity is not evaluable based on the images in the manuscript.

Cell cycle measurements should be presented properly. While PI-staining per se has the danger of counting dead G2/M cells as S or G1 cells, my major issue is the lack of repeats and statistics. Similarly, ex vivo staining (Ki67) needs quantification and statistical analysis.

In the in vivo study, xenograft-allograft discrepancy must be cleared. Also, all cell lines and primary cell cultures should be characterized properly in the methods.

The reviewer has doubts that single-cell suspensions from clinical tumors get back to organoids properly. In practice, it can be a group of cells, even containing non-cancerous cells of the microenvironment, but organoids should carry organ structure, which is not the case after single-cell suspension culturing. Therefore I think it is important to rename this part to primary cancer spheroids. Similarly, term "tissue-derived cells" makes no sense, as all cells are tissue-derived. Importantly, advanced 3D organoids and patient-derived models skip the step of cell dissociation, that is why they are not selecting for quickly proliferating happy-to-be-alone cells.

My detailed comments are in the manuscript pdf. I suggest to avoid any references from the results, as they belong to other chapters (they are not new results).

Mark that the structure (figures are way below the according text) comments are more for the editorial office, I am glad if the Authors can help sign this to the journal.

Overall, with the above changes, I believe that this interesting manuscript has potential to be interesting for the scientific community.

Author Response

(The authors gave the same response as above.)

Reviewer 4 Report

Dear authors,

you find my suggestion in the attached file.

Kind regards

Author Response

(The authors gave the same response as above.)
